# WIN-WIN: TRAINING HIGH-RESOLUTION VISION TRANSFORMERS FROM TWO WINDOWS

**Vincent Leroy, Jerome Revaud, Thomas Lucas & Philippe Weinzaepfel**
Naver Labs Europe
`firstname.lastname@naverlabs.com`

## ABSTRACT

Transformers have become the standard in state-of-the-art vision architectures, achieving impressive performance on both image-level and dense pixelwise tasks. However, training vision transformers for high-resolution pixelwise tasks has a prohibitive cost. Typical solutions boil down to hierarchical architectures, fast and approximate attention, or training on low-resolution crops. This latter solution does not constrain architectural choices, but it leads to a clear performance drop when testing at resolutions significantly higher than that used for training, thus requiring ad-hoc and slow post-processing schemes. In this paper, we propose a novel strategy for efficient training and inference of high-resolution vision transformers. The key principle is to mask out most of the high-resolution inputs during training, keeping only N random windows. This allows the model to learn local interactions between tokens inside each window, and global interactions between tokens from different windows. As a result, the model can directly process the high-resolution input at test time without any special trick. We show that this strategy is effective when using relative positional embedding such as rotary embeddings. It is 4 times faster to train than a full-resolution network, and it is straightforward to use at test time compared to existing approaches. We apply this strategy to three dense prediction tasks with high-resolution data. First, we show on the task of semantic segmentation that a simple setting with 2 windows performs best, hence the name of our method: Win-Win. Second, we confirm this result on the task of monocular depth prediction. Third, to demonstrate the generality of our contribution, we further extend it to the binocular task of optical flow, reaching state-of-the-art performance on the Spring benchmark that contains Full-HD images with an order of magnitude faster inference than the best competitor.

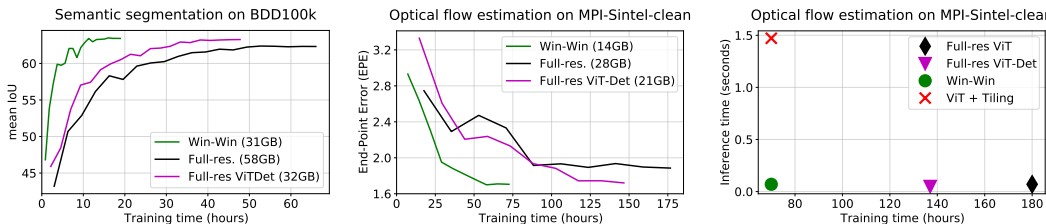

Figure 1: **Validation performance *vs.* training time** on semantic segmentation *(left)* and optical flow *(middle)*. We compare our two-window training (Win-Win) to a standard full-resolution training as well as a sparsification of the attention following ViT-Det (Li et al., 2022a). We indicate the memory usage in parenthesis in the legend. Compared to full-resolution training, Win-Win allows to reduce the training time by a factor 3∼4 and to half the memory usage while reaching a similar performance. **Training and inference times** on optical flow, for Win-Win *vs.* other strategies *(right)*. ViT+Tiling corresponds to a setup similar to CroCo-Flow (Weinzaepfel et al., 2023) where the model is trained on random crops, but requires a tiling strategy at inference. While Win-Win is as fast to train as the latter, it can directly process full-resolution inputs at test time.

Figure 2: **Overview of Win-Win, our approach for high-resolution training of ViTs**. We show that certain masking configurations can generalize to full-resolution at inference time. Specifically, using 2 random squares, which allows to model both local interactions inside each square and global interactions with patches from the other square, is enough. This also offers the advantage of speeding up training and decreasing memory usage considerably, since most image patches are discarded. Our framework is general and applies to binocular tasks as well, *e.g.*, optical flow (see Figure 3).

# 1 INTRODUCTION

We explore the training of *non-hierarchical* vision transformers (Dosovitskiy et al., 2021) (ViTs) for dense tasks at *high resolutions*. Self-attention (Vaswani et al., 2017) based architectures have been shown to scale well with large amounts of data (Dosovitskiy et al., 2021; Touvron et al., 2021; Dehghani et al., 2023a) and can be trained in a self-supervised manner (Caron et al., 2021; Xie et al., 2022; He et al., 2022; Weinzaepfel et al., 2022; Feichtenhofer et al., 2022; Tong et al., 2022). Their generic structure allows them to be finetuned on both image-level and pixel-level downstream tasks (Caron et al., 2021; He et al., 2022; Oquab et al., 2023; Weinzaepfel et al., 2022), however, the applicability of vanilla vision transformers to high-resolution images is limited, due to the quadratic complexity of global self-attention. This problem is compounded by the fact that ViTs tend to generalize poorly to test resolutions significantly higher than that seen at training, which was recently evidenced in several works (Tian et al., 2023; Ali et al., 2021; Oquab et al., 2023; Ranftl et al., 2021).

Existing solutions can broadly be separated into three categories. First, one can replace the quadratic global attention by either a local or windowed attention mechanism (Li et al., 2022a; Liu et al., 2021; 2022a), or by using an approximation of the attention with sub-quadratic complexity (Ali et al., 2021; Wang et al., 2020a). The second category of approach aims at reducing the number of tokens by using hierarchical representations with pooling, downsampling or pruning (Wang et al., 2021; Fan et al., 2021; Li et al., 2022b; Zhang et al., 2021; Liu et al., 2021; Yin et al., 2022; Pan et al., 2021; Chen et al., 2023; Rao et al., 2021), or by varying the patch resolution (Beyer et al., 2023; Lin et al., 2023). This is typically effective for tasks that do not require pixelwise predictions. The third category consists in keeping a vanilla ViT architecture and training on small-sized crops of fixed size, while resorting to tiling at test time to obtain high-resolution outputs (Weinzaepfel et al., 2023; Huang et al., 2022; Shi et al., 2023; Jaegle et al., 2022), *i.e.* with a sliding window strategy, where multiple predictions from overlapping crops can be aggregated. An alternative consists in performing most training at lower resolutions and finally finetuning at the target high resolution (Zheng et al., 2021; Ranftl et al., 2021), but this remains very costly. In this paper, we propose a novel path that maintains *vanilla* self-attention, uses a *non-hierarchical* transformer backbone, does not require intensive fine-tuning, and produces high-resolution outputs in a *single forward pass* at test time.

Our proposed approach to train a high-resolution transformer with vanilla self-attention relies on masking most of the input tokens, leading to a $3-4\times$ *faster training and* $2\times$ *reduced memory*, see Figure 1. Our main contribution is to show that *specific masking configurations*, unlike the random one typically employed in masked input modeling (He et al., 2022; Tong et al., 2022), must be used to enable high-resolution generalization at inference time, see Figure 2. These configurations allow to jointly consider global and local interactions, which is key for dense prediction tasks. Our proposed solution is to randomly select multiple windows in the input image, so that both global (inter-windows) and local (intra-windows) interactions occur during training. We restrict our study to rectangular windows that naturally lends themselves to the convolutional heads typically used with transformers backbones for dense prediction tasks (Ranftl et al., 2021; Liu et al., 2022b). Empirically, we find that a simple setup with two squared windows of the same shape performs as well as more elaborate strategies. We therefore call our training strategy with two windows *Win-Win*.

To demonstrate the generality of our training framework, we show its effectiveness on monocular and binocular tasks. Specifically, we experiment with semantic segmentation, monocular depth and optical flow estimation. On the first two tasks, we achieve a final performance on par with more

elaborate training strategies that can require test-time processing tricks such as sliding window, which is slow and produces artifacts. Applying this idea to the last task of optical flow estimation is non trivial as it involves an additional input image for which the window sampling strategy depends of the first image's windows. We thus propose an extension of our window sampling strategy to the case of multiple image. Using it, we obtain state-of-the-art performance on the Full-HD Spring benchmark (Mehl et al., 2023) without employing task-specific designs nor requiring tiling at test time.

Our training strategy allows to test directly at the target resolution, however the statistics of the distribution of self-attention token similarities change when more tokens become visible at test time. We thus compensate for this using a temperature factor in the softmax of the attention, validated using the performance on the full-resolution training images.

## 2 RELATED WORK

**High-Resolution ViTs** are costly to train due to the quadratic complexity of the global attention mechanism with respect to the number of input tokens. A common strategy is to train on a small resolution and test on higher-resolution inputs (Ranftl et al., 2021; Oquab et al., 2023; Zhang et al., 2021), or to use a fixed scale during both training and test (Li et al., 2022a; Kirillov et al., 2023). In the first case, it is consistently reported that the train/test discrepancy consistently decreases performance. For instance, DPT (Ranftl et al., 2021) trains on $480\times480$ crops and notices a clear drop in performance as test resolution increases. Three trends emerged to decrease training cost: subquadratic approximations of the global attention, local/sparse attention and hierarchical approaches.

In more details, the first strategy is to approximate the original global attention mechanism with sub-quadratic variants. The most computationally efficient methods have a linear complexity w.r.t. to the number of tokens, via an approximations of the softmax (Choromanski et al., 2021), by spatial reduction (Wang et al., 2020a; 2021) or linearization (Ali et al., 2021) of the query-key product.

Secondly, instead of modifying the quadratic attention, several works propose to sparsify the affinity matrix between queries and keys, *e.g.* attention sparsification. This can either be done using local attention mechanisms (Vaswani et al., 2017; 2021), via subsampling (Wu et al., 2021; Zhang et al., 2022; Huang et al., 2019) or a combination of local and low-resolution global attention (Zhang et al., 2021; Chu et al., 2021). For instance, ViT-Det (Li et al., 2022a) proposes to use a windowed attention in almost all transformer blocks except 4 of them, in order to still model global interactions for object detection. SAM (Kirillov et al., 2023) uses a similar backbone for segmentation. However, this remains costly to train (only 25% training speed-up, see Figure 1) and tends to achieve slightly lower performance than using vanilla attention, as shown in (Li et al., 2022a).

Thirdly, several methods have proposed to reduce the number of tokens (Lin et al., 2023), possibly with the use of hierarchical structures, either regular (Liu et al., 2021; 2022a; Wu et al., 2021; Fan et al., 2021; Li et al., 2022b; Wang et al., 2021; Chen et al., 2022b) or irregular (Yuan et al., 2021; Bolya et al., 2023; Xu et al., 2022). However, this redesigning of the original ViT does not allow to easily leverage task-agnostic large-scale pre-training, which is generally performed with a standard ViT backbone. Finally, another solution is to drop random tokens as proposed in (Fang et al., 2023) for object detection, a task that does not require a pixelwise prediction, or more recently in a concurrent work (Dehghani et al., 2023b) for image classification where the goal is mainly to allow batching despite various image sizes during training. However, when considering pixelwise prediction tasks, reducing the number of tokens might harm the final performance, given the importance of low-level details for these tasks.

Compared to all these approaches, Win-Win allows to train a general model from a structured subset of tokens, and thus to keep the original global attention in all blocks. It has the same cost as when training from small crops while enabling full-resolution testing, without resizing nor tiling.

**Dense Binocular Tasks.** Even though the quadratic complexity is a problem common to all methods in optical flow, we only focus here on methods that leverage transformers for this task; we refer the reader to (Zhai et al., 2021) for a more general review. In particular, we study the state of the art through the scope of the computational burden of high-resolution training and testing. Most of the previous art devised task-specific architectures (Huang et al., 2022; Li et al., 2021; Sui et al., 2022; Zhao et al., 2022; Xu et al., 2023). These methods all make use of a dedicated decoder that

takes as input the query image and a learned representation. This representation, be it a 4D cost-volume (Huang et al., 2022; Sui et al., 2022), the result of pure cross-attention (Zhao et al., 2022; Xu et al., 2023) or a hybrid of both (Li et al., 2021), is where the computational complexity lies. The vanilla global attention of transformers is often replaced with either coarse-to-fine approaches (Zhao et al., 2022; Xu et al., 2023) or through attention sparsification (Li et al., 2021). When it is not, tiling is employed at test-time (Huang et al., 2022; Shi et al., 2023; Jaegle et al., 2022). In (Li et al., 2021), gradient checkpointing (Griewank & Walther, 2000) is also used, which complexifies the procedure. In contrast, the philosophy of this work lies in the idea that a simpler general architecture could be used without bells and whistles. Some previous works explore this direction (Weinzaepfel et al., 2023; Jaegle et al., 2022), but still train on low resolutions and resort to tiling at test-time to alleviate the training computational burden. To the best of our knowledge, we are the first to show that testing directly at high resolution is possible while training with a reasonable cost.

## 3 TRAINING FROM MULTIPLE WINDOWS: WIN-WIN

The standard vision transformer (Dosovitskiy et al., 2021) views an input image $x$ as a set $\mathcal{P}$ of patches, or equivalently, tokens. A series of blocks alternating multi-head self-attention and a multi-layer perceptron (MLP) then processes this set of tokens. Win-Win relies on the idea of using a subset $\mathcal{P}' \subset \mathcal{P}$ during training, *i.e.* masking (or rather, discarding) the other tokens, while allowing to directly process $\mathcal{P}$ at test time. The training complexity is thus reduced from $\mathcal{O}(|\mathcal{P}|^2)$ to $\mathcal{O}(|\mathcal{P}'|^2)$; the size of subset $\mathcal{P}'$ can be adapted to the memory budget and made independent from the input image resolution.

Given how local interactions are important for vision tasks (Chen et al., 2022a; Wang et al., 2023), it seems desirable to preserve as many *neighboring tokens* as possible in $\mathcal{P}'$. This can be easily implemented, *e.g.* by defining the selected tokens to be inside a rectangular crop. On the other hand, it seems crucial to preserve long-range interactions as well. Without them, generalizing to high-resolution images might be impossible due to the domain gap, since long-range dependencies would have never been observed in the small training crops. Such drop of performance when increasing the resolution was recently evidenced in (Tian et al., 2023; Ali et al., 2021; Oquab et al., 2023; Ranftl et al., 2021). The main idea of Win-Win thus consists in selecting the tokens of $\mathcal{P}'$ in a structured configuration, where both local and long-range token interactions are guaranteed to be present. Specifically, we select tokens from a set of $N \geq 2$ non-overlapping rectangles $\{R_i\}_{i=1..N}$, see Figure 2:

$$\mathcal{P}' = \{p \mid \exists i \in \{1..N\}, p \in R_i\}. \tag{1}$$

We now detail the token selection procedure, architectural details and generalization to binocular tasks of our Win-Win approach.

**Window distribution.** The principle of Win-Win is to randomly sample $N$ non-overlapping windows for each training image. Note that, thanks to randomness, all training token positions end up being selected at some point during training. Experimentally, one of our findings is that the simplest strategy (*i.e.*, sampling $N = 2$ squared windows of the same size) performs best, see Section 4. Beyond choosing $N$, different window sizes can be chosen, depending on the compute budget desired for training.

**Convolutional heads.** State-of-the-art ViT-based architectures for pixelwise prediction tasks typically rely on convolutional heads (Ranftl et al., 2021; Liu et al., 2022b; Weinzaepfel et al., 2023). In contrast to unstructured MAE-like random masking (He et al., 2022; Fang et al., 2023; Dehghani et al., 2023b), Win-Win is compatible with a convolutional head. Token features output by the transformer backbone can be split and reshaped into features maps from the $N$ rectangles, to which convolutions can be applied separately as illustrated in Figure 2.

**Positional embeddings.** Dense tasks such as semantic segmentation are typically translation equivariant, which becomes a useful guiding principle when designing deep models. Classical absolute positional embeddings that are added to the signal, either learned (Dosovitskiy et al., 2021) or using cosine functions (Vaswani et al., 2017), do not satisfy this property. We therefore employ relative positional embeddings that are applied directly at the level of self-attention computations. They can either be learned parameters (Liu et al., 2022a; Li et al., 2022b), outputs of a neural network (Liu et al., 2022a) or given by transforms only applied to queries and keys (Su et al., 2021). In this work,

Figure 3: **Overview of Win-Win for the task of optical flow estimation.** Masking is performed asymmetrically so that selected windows in the second frame are more likely to correspond to windows randomly selected in the first frame. The rest of the framework remains identical.

we use this latter option as it does not involve any learnable parameters, and empirically performs better than absolute embeddings.

**Test time.** At test time, the windowed sampling scheme can simply be removed and the full image, *i.e.* the full set of tokens $\mathcal{P}$ is processed. Note that memory requirement at test time are drastically lower, since intermediate tensors can be immediately freed during inference.

**Feature distribution changes.** Using the full image at test time induces a change in the number of tokens compared to training. Although self-attention handles arbitrary numbers of tokens, the softmax distribution can be altered by this increase. To compensate for this, we tune the temperature hyper-parameter in the softmax attention$_\tau(Q, K, V) = \text{softmax}\left(\frac{1}{\tau}QK^T\right)V$, which normally defaults to $\tau = \sqrt{d}$, where $d$ is the feature dimension of each head. Once Win-Win training is performed, we validate $\tau$ on full-resolution images of the train set (*i.e.* without masking). Please refer to Appendix E for more details.

**Generalization to binocular tasks.** Win-Win is a general framework for transformer-based architectures that can also be applied to multi-image tasks. For instance, and without loss of generality, we focus in this paper on the binocular task of optical flow estimation. The task consists of predicting the displacements of pixels between two consecutive video frames. In this scenario, masking only one frame is not sufficient to limit the training complexity, and we thus need to mask both frames simultaneously, see Figure 3. This is not trivial, since valuable information with respect to a given window in the first frame is located at a particular spot in the second frame. Hence, simply applying the random crop strategy on each frame *independently* would likely lead to a very sparse training signal, as matching pixels would have a low chance of being visible in both inputs.

To circumvent this issue, we propose a simple binocular extension to Win-Win. We first sample $N$ non-overlapping random windows in the first frame. We then evaluate whether each token in the second frame has a corresponding visible token in the first frame, based on the ground-truth flow. We finally sample $M$ non-overlapping windows in the second frame, this time using random sampling weighted by the amount of matched tokens inside each window. We refer to Appendix B for more details. We experiment in Section 4.2 with this strategy as well as with simpler strategies (*e.g.* using the same windows in both frames) and demonstrate superior performance for the proposed sampling scheme.

## 4 EXPERIMENTS

In this section, we first validate our Win-Win training strategy on a monocular task (semantic segmentation) in Section 4.1 and then present results for the binocular task of optical flow (Section 4.2). Please refer to Appendix D for more results on the monocular depth estimation task.

### 4.1 MONOCULAR TASK RESULTS

**Experimental setup.** Experiments are performed on the BDD-100k dataset (Yu et al., 2020) that comprise 7,000 training images and 1,000 validation images in a driving scenario with 19 semantic classes. All images have a relatively high resolution of 1280×720 pixels. We report the mean intersection-over-union (mIoU) metric on the validation set in percentage.

**Training details.** We use a ViT-Base encoder (Dosovitskiy et al., 2021) attached to a Conv-NeXt head (Liu et al., 2022b; Bachmann et al., 2022), finetuned for 100 epochs from a CroCo pretrained

Table 1: **Semantic segmentation ablations** with the mIoU↑. *Left:* impact of the number of windows for a target number of tokens of 1024, with and without the softmax temperature. *Right:* impact of different rectangle shapes with 2 windows.

| windows | (# tokens) | w/o temp. | w/ temp. |
|---|---|---|---|
| 1 win. 32×32 | (1024) | 61.3 | 61.8 |
| 2 win. 22×22 | (968) | **63.6** | **63.6** |
| 3 win. 18×18 | (972) | 63.0 | 63.2 |
| 4 win. 16×16 | (1024) | 63.0 | 63.2 |
| 6 win. 13×13 | (1014) | 62.3 | 62.5 |

| 2 windows | (# tokens) | w/ temp. |
|---|---|---|
| 22×22 + 22×22 | (968) | 63.6 |
| 26×19 + 26×19 | (988) | 62.6 |
| 26×26 + 19×19 | (1037) | 63.4 |
| 28×28 + 15×15 | (1009) | **63.7** |
| 30×30 + 11×11 | (1021) | 63.1 |
| 22×22 + 12×12 | (628) | 62.8 |

model (Weinzaepfel et al., 2022; 2023) with RoPE positional embeddings (Su et al., 2021), except otherwise stated. Detailed training settings are in Appendix F.

**Training with multiple rectangles.** Table 1 (left) reports results obtained with our multi-window training strategy, for a fixed budget of approximately 1024 tokens (up to rounding errors depending on resolutions) out of 3600 tokens in the full-resolution input, with varying numbers of windows. A single window yields the worst performance, demonstrating the importance of learning long-range interactions. Using more windows performs better, and we find that 2 windows is enough to achieve the best performance.

**Impact of softmax temperature.** Table 1 (left) shows the results obtained with and without a temperature parameter added to the softmax, to account for the discrepancy of the number of tokens during training (around 1024) and testing (around 3600). The impact is moderate, with results improved in terms of mIoU by a margin between 0 and 0.5, showing near-perfect generalization capabilities when using two windows.

**Impact of positional embeddings.** Next, we compare our strategy when using standard cosine absolute positional embeddings instead of RoPE. The mIoU goes down from 63.6% to 57.0%. This clear drop of performance can be explained by the fact that absolute positional embeddings suffer from interpolation from the low-resolution pre-trained models and to the absence of translation equivariance. We also find that trying to apply a correction to the temperatures in the softmax did not help either in this setting (-7% mIoU). We hypothesize that this is caused by the relationship between the softmax temperature $\tau$ and the self-probability in the attention when using relative positional embedding, see Appendix E.

**Window sizes.** Fixing the number of windows to $N=2$, we vary the window sizes in Table 1 (right). We observe similar performance (between 63 and 64 mIoU) as long as the number of tokens remains fixed at ∼1024. Square windows of identical sizes thus suffice, and we keep this setting (first row) as default setting for the rest of the experiments for simplicity. We also try to reduce the size of the second window, but this result in a small drop of performance (last row of Table 1). Finally, we experiment with (a) adding other isolated random tokens, in addition to the windows or (b) choosing window sizes randomly at each iteration, but we do not notice any improvement, see Appendix C for details. Overall, we note that Win-Win is robust to various window designs, and that simpler (two squared windows of the same size) is better.

**Comparison to other baselines.** We compare Win-Win to other baselines in Table 2 (left). The first baseline consists of training at full resolution, denoted as 'Full-res ViT', but has a large cost due to 3600 tokens going into quadratic self-attention blocks (see also Figure 1 left). To mitigate this, ViT-Det (Li et al., 2022a) and SAM (Kirillov et al., 2023) have proposed to replace global attention by windowed attention in the ViT, except at 4 layers regularly spread across blocks where global attention is kept. While altering the attention mechanism, this alternative (denoted as 'ViT-Det') performs roughly on par with Full-res. However, training time is only reduced by about 25%, see Figure 1. In comparison, Win-Win trained on 2 windows of size 352×352 (*i.e.* 22×22 tokens) slightly improves the performance while reducing the training time by a factor of 4 and the memory requirement by a factor of 2. We hypothesize that the slight improvement in performance is due to the masking acting as data augmentation for this semantic task (Chen et al., 2020).

Table 2: **Semantic segmentation results.** *Left:* comparison to other train/test setup (training window size in tokens, test resolution in pixels). *Right:* results with varying number of tokens for Win-Win with 2-windows or training with 1 crop and testing with tiles of the same crop size.

| Training | Train Tokens | Test Resolution | mIoU↑ |
|---|---|---|---|
| Full-res. ViT | 3600 | 1280×720 | 63.2 |
| Full-res. ViT-Det | 3600 | 1280×720 | 63.3 |
| Random tokens | 968 | 1280×720 | 59.9 |
| crop 32×32 | 1024 | 1280×720 | 61.8 |
| crop 32×32 | 1024 | resize 512×512 | 56.1 |
| crop 32×32 | 1024 | tiling 512×512 | 62.6 |
| **Win-Win (2-win. 22×22)** | 968 | 1280×720 | **63.6** |

| windows | (# tokens) | mIoU↑ |
|---|---|---|
| crop 32×32 | (1024) | 62.6 |
| 2-win. 22×22 | (968) | **63.6** |
| crop 24×24 | (576) | 62.6 |
| 2-win. 17×17 | (578) | **63.1** |
| crop 14×14 | (196) | **59.4** |
| 2-win. 10×10 | (200) | **59.4** |

We also compare to a baseline with randomly masked tokens (*e.g.* like MAE He et al. (2022)) which we clearly outperform. This suggests that our method allows to better leverage local interactions inside each window. Additionally, our approach enables to train a convolutional head on top, even if this does not explain alone the gains (Appendix C). We finally compare to a baseline that is trained on a single fixed-size crop but for which we evaluate three different test-time strategies: either (i) perform full-resolution inference without any change, (ii) downscale the test image and upscale the output prediction (denoted as *resize*), or (iii) split the image into a set of (overlapping) fixed-size crops and aggregate per-crop predictions afterwards (denoted as *tiling*). Note that tiling requires many forward passes. For instance, up to 6 predictions per pixel are computed to go from (512×512) to (1280×720) with a crop overlap of 50%, and up to 33 predictions per pixel with 90% overlap. We find that tiling overall yields the best baseline results, most likely due to the fact that the first two baselines result in large domain gaps *w.r.t.* training data. To conclude, our proposed multi-window training strategy achieves *the best of both worlds*: it allows to directly process high-resolution images at test time in a single forward pass, without any sliding window or other strategy required, while being cheaper to train than full-resolution approaches.

**Impact of the number of tokens.** In Table 2 (right), we experiment with different numbers of training tokens (200, 580, 1024) and compare the performance of Win-Win *w.r.t.* training with a single crop and using tiling at test time. When using 200 tokens, Win-Win performs on par with the baseline but allows to produce high-resolution results at test time in one forward pass instead of using tiling. With more tokens, Win-Win even outperforms the single-crop tiling baseline.

## 4.2 BINOCULAR TASK RESULT

We now experiment with the binocular task of optical flow estimation. We first select the window sampling strategy on a small synthetic dataset built for this purpose, and then evaluate the best configurations on the MPI-Sintel (Butler et al., 2012) and Spring (Mehl et al., 2023) benchmarks.

**Experimental setup.** We base our model on CroCo-Flow-Base (Weinzaepfel et al., 2023). It comprises a Siamese ViT-Base image encoder followed by a ViT-Base decoder with cross-attention to exchange information with the second frame, and a DPT head (Ranftl et al., 2021). We refer to Appendix F for more details.

**Multi-Window training for optical flow.** Our masking strategy (Section 3) for optical flow consists of extracting $N$ and $M$ windows in the first and second frame, respectively. We evaluate different options for sampling $N$ and $M$ in Table 3. Results are reported on a small synthetic dataset, constructed akin to AutoFlow (Sun et al., 2021), with a smaller network architecture for the sake of speed. The performance improves significantly when going from $M=2$ to $M=3$ windows, and using $M\geq3$ windows performs similarly. Stochasticity denotes the level of randomness when sampling windows in the second frame. No

Table 3: **Comparison of different window sampling strategies** for optical flow on a synthetic test dataset.

| Frame 1 ($N$) windows | Frame 2 ($M$) windows | Stochasticity | EPE↓ |
|---|---|---|---|
| 2 win. 10×10 | 2 win. 10×10 | 0.2 | 1.92 |
| | 3 win. 8×8 | | 1.24 |
| | 4 win. 7×7 | | **1.17** |
| | 5 win. 6×6 | | 1.20 |
| 2 win. 10×10 | 4 win. 7×7 | 0.3 | **0.96** |
| | | 0.4 | 0.99 |
| 3 win. 8×8 | 3 win. 8×8 | 0.2 | 1.43 |
| | 3 win. 7×7 | | 2.81 |
| 1 win. 14×14 | same win. | - | 7.32 |
| 2 win. 10×10 | | | 2.70 |

Table 4: **Optical flow results on MPI-Sintel validation set.** *Left:* Win-Win with different number of tokens. *Right:* comparison between full-resolution training (with full self-attention or ViT-Det), Win-Win (2 windows in the first image, 4 in the second) and a baseline trained on crops.

| Window scheme | # Tokens (img1/img2) | val EPE↓ |
|---|---|---|
| 2 win. 10x10, 4 win. 7x7 | (200/196) | 2.00 |
| 2 win. 12x12, 4 win. 9x8 | (288/288) | 1.96 |
| 2 win. 14x14, 4 win. 10x10 | (392/400) | 1.83 |
| 2 win. 16x16, 4 win. 11x11 | (512/484) | **1.67** |

| Training | # Tokens (img1/img2) | Test Resolution | val EPE↓ |
|---|---|---|---|
| Full res. ViT | (1728/1728) | 1024×432 | 1.79 |
| Full res. ViT-Det | (1728/1728) | 1024×432 | 1.69 |
| Crop 20×24 tokens | (480/480) | 1024×432 | 91.31 |
| Crop 20×24 tokens | (480/480) | Resize 384×320 | 2.44 |
| Crop 20×24 tokens | (480/480) | Tiling 384×320 | 1.77 |
| **Win-Win (2 win. 16×16, 4 win. 11×11)** | (512/484) | 1024×432 | **1.67** |

stochasticity means that the selection is optimal *w.r.t.* windows selected in the first frame. We experiment with different amounts of stochasticity in the window selection process and obtain the best performance with a value of 0.3. Using $N=3$ windows in the first frame tends to degrade results slightly, though the training still works and seems robust to this change. As a sanity check, we also compare to a simpler variant where the same windows are used in the second frame and obtain a degraded performance. Finally, we also compare in this setting to the case of a single window, which significantly degrades performance due the lack of global interaction modeling.

**Results on MPI-Sintel validation.** Using the best settings found previously (*i.e.* $N=2$ and $M=4$), we evaluate Win-Win on MPI-Sintel (Butler et al., 2012) in Table 4. Models are trained on FlyingChairs (Dosovitskiy et al., 2015), FlyingThings (Mayer et al., 2016), and MPI-Sintel from which we keep two sequences apart for validation. We report the average endpoint error (EPE) on this validation set in the 'clean' rendering. On the left, we first compare performance for different number of training tokens (200, 300, 400 and 500) for Win-Win while keeping the window scheme (number of windows in each image) constant. The performance improves when the number of visible tokens grows, and we thus use the 500 tokens setting, a similar value as used in CroCo-Flow (Weinzaepfel et al., 2023). On the right, we compare Win-Win to full-resolution training (with or without ViT-Det). This has a higher cost for training, see Figure 1 middle. Win-Win obtains a slightly better performance while significantly reducing the training cost.

We also compare to several baselines trained on fixed-size crops of resolution 384×320 (480 tokens) in Table 4. Applying directly the model to full-resolution test frames performs extremely poorly. The second test strategy, consisting of downscaling the test frame to the training crop size, achieves better results, but still relatively poor due to the loss of details when downscaling the frames and upscaling the predicted flow. This again highlights that ViTs suffer from train/test resolution discrepancy. The best inference strategy for this baseline is obtained using a tiling-based approach that requires many forward passes at test time, leading to a high inference cost, see Figure 1 (*right*). In contrast, Win-Win (last row) achieves the best EPE overall while predicting directly at the full resolution.

Table 5: **Comparison to the state of the art on the MPI-Sintel benchmark** on the clean and final renderings. [†] means that tiling is used at test-time. Win-Win has a generic architecture (ViT-based without cost volume, etc.) and uses a single forward pass.

| Method | test EPE↓ Clean | Final |
|---|---|---|
| PWC-Net+ (Sun et al., 2019) | 3.45 | 4.60 |
| RAFT (Teed & Deng, 2020) | 1.61 | 2.86 |
| CRAFT (Sui et al., 2022) | 1.44 | 2.42 |
| FlowFormer[†] (Huang et al., 2022) | 1.16 | _2.09_ |
| FlowFormer++[†] (Shi et al., 2023) | _1.07_ | **1.94** |
| SKFlow (Sun et al., 2022) | 1.30 | _2.26_ |
| GMFlow+ (Xu et al., 2023) | **1.03** | 2.37 |
| CroCoFlow[†] (Weinzaepfel et al., 2023) | 1.09 | 2.44 |
| **Win-Win** | 1.15 | 2.34 |

**Comparison to the state of the art.** We finally compare to the state of the art on two benchmarks. First, we evaluate on MPI-Sintel in Table 5. In this setup, we finetune our model using

Table 6: **Comparison to the state of the art on the Spring benchmark** with the number of outliers (error over 1px) as well as the endpoint error (EPE) over all pixels, or over pixels with flow norm in [0,10] (s0-10), in [10,40] (s10-40) and over 40 pixels (s40+). [†] means that tiling is used at test time. [‡] means methods submitted by the leader-board's authors.

| Method | 1px↓ | EPE↓ | s0-10↓ | s10-40↓ | s40+↓ |
|---|---|---|---|---|---|
| FlowFormer[†‡] (Huang et al., 2022) | 6.510 | 0.723 | 0.217 | 0.429 | 5.753 |
| MS-Raft+[‡] (Jahedi et al., 2022) | 5.724 | 0.643 | 0.154 | 0.398 | 5.403 |
| CroCo-Flow[†] (Weinzaepfel et al., 2023) | **4.565** | 0.498 | **0.126** | **0.338** | 4.046 |
| **Win-Win** | 5.371 | **0.475** | 0.129 | 0.375 | **3.639** |

| Reference frame | Win-Win | Error for Win-Win | Error for CroCo-Flow |
|---|---|---|---|

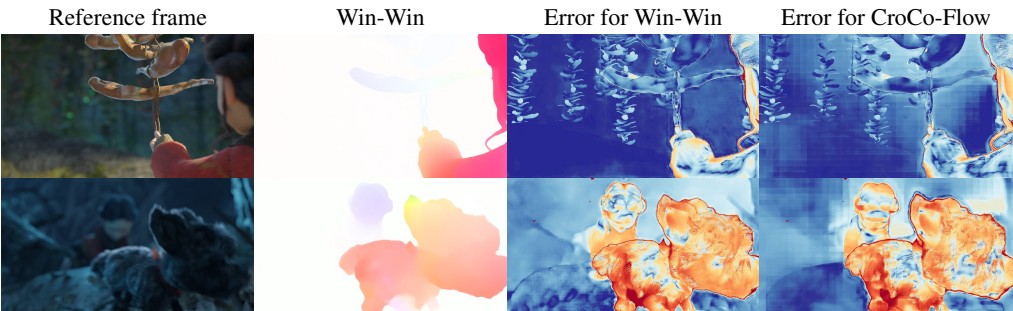

Figure 4: **Example results on Spring** (test set). For the error maps, blue and red denote low and high errors, respectively. *Block artefacts* are clearly visible for CroCo-Flow but absent for Win-Win.

TartanAir (Wang et al., 2020b) in addition to FlyingChairs, FlyingThings and MPI-Sintel, following CroCo-Flow (Weinzaepfel et al., 2023). We finetune the softmax temperature $\tau$ in each attention head for half an hour on full-resolution images from the train set while freezing the network weights. Doing so yields a performance improvement of about 0.05 in EPE. Our model performs closely to the other transformer-based models such as CroCoFlow (Weinzaepfel et al., 2023), FlowFormer (Huang et al., 2022) and FlowFormer++ (Shi et al., 2023). Note that Win-Win is based on the same architecture as CroCo-Flow, except that the latter is using a ViT-Large encoder backbone (we use a ViT-Base), yet Win-Win is still competitive. Furthermore, all these methods rely on tiling at test time and thus require multiple forward passes per frame pairs, with ad-hoc designs to merge the overlapping predictions. In contrast our method is simple and fast at test time, since predictions are made directly from the high-resolution inputs, see Figure 1 (right). Our method performs also close to highly task-specific approaches such as GMFlow+ (Zhao et al., 2022) which rely on coarse-to-fine schemes and cost volumes, whereas we use generic vision transformers.

Results for the more recent Spring benchmark, with $1920 \times 1080$ Full-HD images, are reported in Table 6. On these larger images, it is even more critical to reduce training and inference costs, compared to full-resolution training and to tiling-based approaches. We finetune Win-Win on this dataset using 2025 tokens (*i.e.* 25% of 8100 tokens at full-resolution, which is a comparable proportion than for MPI-Sintel) for 16 epochs. Win-Win yields state-of-the-art performance for the EPE metric, particularly improving with large displacements, and even beating CroCo-Flow (Weinzaepfel et al., 2023) that uses a $3 \times$ larger backbone. Moreover, the usage of tiling by CroCo-Flow at inference results in strong blocking artifacts as illustrated in Figure 4 (right). In comparison, Win-Win can directly process the Full-HD frames at test time and yields smooth predictions without any artifacts. Again, inference with Win-Win is more than an order of magnitude faster than that of CroCo-Flow.

## 5 CONCLUSION

We have shown for the first time that high-resolution vision transformers can be efficiently trained with a multi-window training strategy and directly applied to the target resolution at test time. In other words, Win-Win combines the benefits of both (1) reduced training cost as training from crops, and (2) a direct inference as when training in high-resolution.

**Reproducibility statement.** The training setups are described in the experimental section (Section 4), and we included hyper-parameters details in Appendix F. Thanks to the simplicity of our approach, we believe Win-Win to be easily reproducible in other training settings.

**Ethics statement.** We have read the ICLR Code of Ethics and ensures that this work follows it. Datasets used are all publicly available. We believe the negative impacts of this work to be rather limited. In fact, the application of our method does not allow for new tasks or new behaviors than that of previous works, but rather eases the training and lowers the costs. The societal impacts are the same than that of ViTs.

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

APPENDIX

This appendix presents detailed explanations and additional experiments mentioned in the main paper. Namely, we first study the robustness to test resolution in Appendix A and provide in Appendix B an in-depth explanation of the flow-guided multi-window selection procedure. Next, we elaborate on other window strategies in Appendix C, including using random tokens as windows. We then present some results on monocular depth estimation in Appendix D and discuss in Appendix E the temperature adjustment in the attention *softmax* used at test-time. We then provide in Appendix F the training setups and compute details, and in Appendix G the study of the variance of the runs in our experiments.

## A    ROBUSTNESS TO TEST RESOLUTION

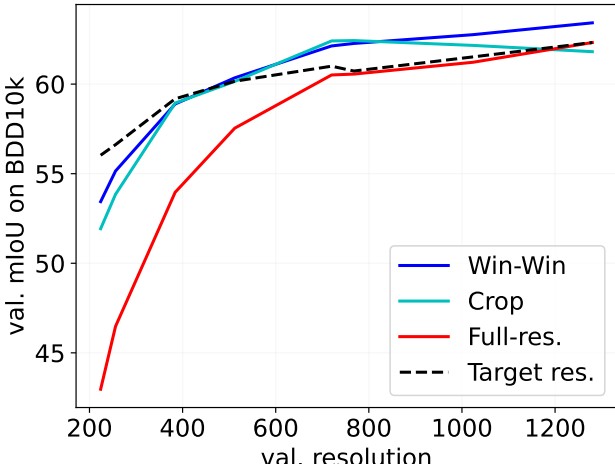

Figure 5: **Robustness to test resolution.** We compare the performance of a model trained with Win-Win, on crops, and at full-resolution when varying the test resolution. Some performance decrease at lower resolution can be explained by the smaller context available, this is why we show in a dashed line the performance when training at the target resolution.

So far, all test images were assumed to have the same resolution known during training. For the case of semantic segmentation, we study the robustness to various test resolutions. To do so, when testing at a given resolution, we sample crops of this given size and evaluate on them. We report results in Figure 5. We observe a clear drop for the full-resolution training. The method trained on $512{\times}512$ crops have drops both for small sized regions and larger regions. These two results highlight again that ViTs are not robust to change of resolution between train and test. Our Win-Win training is more robust to changes of resolution but still drops. However, this drop can be mainly explained by the fact that smaller resolutions allow to leverage less context for predictions. To better measure that, we show in a dashed line a model trained on the target test resolution, which shows a similar drop as for Win-Win.

## B    GUIDING THE WINDOWS IN THE SECOND FRAME WITH THE FLOW

Guiding the windows of the second view purely based on the optical flow would provide the network with part of the answer to solve the training task. On the other hand, randomly selecting windows would lead to a very scarce supervision signal. To overcome this, we guide the windows of the second frame using a perturbed flow. After having chosen a set of rectangular windows in the first frame, the window selection process for the second frame consists in three distinct steps: 1) we bin the visible endpoints of the optical flow for each token in the second frame, 2) we perturb the binning with Gaussian noise, 3) we use a greedy algorithm to sequentially select $N$ rectangular windows on the maximum binned values. A visual explanation is provided in Figure 6, for the two windows shown on the top of the middle column. We compute the binned forward flow, by counting

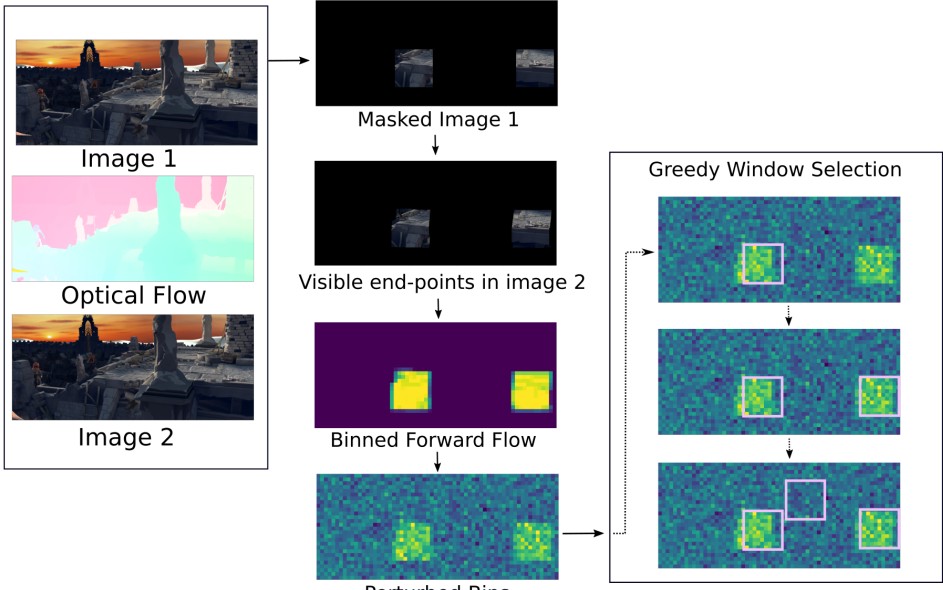

Figure 6: **Overview of the window sampling strategy for the second frame** with three rectangular masks (*pink*) guided by a perturbed optical flow.

the number of flow values that fall inside each token. We then perturb the bin values with Gaussian noise, centered on the mean of the binned flow values. Finally, a greedy algorithm sequentially selects the target number of rectangular windows (here, three) that encompass the maximal values of the perturbed binned forward flow. The only hyperparameter is the standard deviation (*std*) of the normal distribution, expressed as a ratio of the *std* of the binned map.

## C  OTHER WINDOW SETTINGS

**Variable window sizes.** In Table 1 of the main paper, we study the impact of the number of squared windows and also try non-squared windows or windows of different sizes; we have also tried some other variants for which we report results in Table 7. In the left table, we randomize a) the number of tokens used at each training iteration (either fixed to 1024 or randomly chosen in [768,1280]), b) the number of windows (either fixed to 2 or randomly chosen among 2, 3 or 4), c) the ratio of tokens of the respective size of the windows (either of equal size or such that the size of the largest rectangle is up to 2 times bigger than the average), and d) the aspect ratio of the windows (either fixed to squared windows or with an aspect ratio randomly chosen in the range [1/2,2/1]). We observe that all these variants perform on par with the simple setup of two squared windows of constant size.

Table 7: **Semantic segmentation results with other window schemes (randomized windows or extra tokens in addition to the windows).** *Left:* We compare our default setup with two squared windows of 22×22 tokens to some randomized windows scheme, *i.e.*, that changes at every training iteration, either in terms of number of tokens, number of windows, ratio of tokens between the different windows or aspect ratio of the windows. *Right:* impact of adding to the windows sampled at each iteration some random isolated tokens.

| #Tokens | #Windows | Token Ratio | Aspect Ratio | mIoU↑ |
|---|---|---|---|---|
| 968 | 2 | 1 | 1 | 63.6 |
| 1024 | 2 | [0,2] | [1/2,2/1] | 63.5 |
| 1024 | {2,3,4} | [0,2] | [1/2,2/1] | 63.5 |
| [768,1280] | 2 | 1 | 1 | 63.6 |
| [768,1280] | {2,3,4} | [0,2] | [1/2,2/1] | 63.6 |

| Extra Tokens | mIoU↑ | |
|---|---|---|
| | 1 win. 32×32 | 2 win. 22×22 |
| 0 | 61.8 | 63.6 |
| 20 | 61.8 | 63.3 |
| 50 | 61.8 | 63.2 |
| 100 | 61.8 | 63.0 |

**Additional isolated tokens.** Next, we experiment with a combination of 1 or 2 large squared windows (*e.g.* one 32x32 or two 22x22 token windows) and a small set of additional isolated tokens (*i.e.* outside of the windows). While large windows are beneficial to model local interactions and to train the convolutional head, the extra tokens could help to model long-range interactions. Note that these extra tokens are dropped for the convolutional head, and the loss is only applied on the main windows. We report results in the right-hand side of Table 7. We observe that these extra tokens have no impact with one single window, and tend to degrade performance when we use two windows, meaning that the 2-window case is sufficient to learn long-range interactions.

**Additional random tokens.** We also experiment with isolated tokens randomly chosen, *i.e.* we select a large set of windows of size $1 \times 1$, as popularized in Masked Image Modeling (MAE) (He et al., 2022; Xie et al., 2022), instead of rectangular windows. We perform an experimental study on the optical flow estimation task, using the synthetic flow dataset (see Section 4.2 of the main paper). In this case, visible tokens are selected randomly and independently in the first frame. In the second frame, we still use the same flow-based voting mechanism, disturbed by an optional noise to add more diversity. Note that we cannot train a convolutional head in this setting due to the *a-trous* structure of the output (only a few tokens/patches get an output), which is a severe limitation. We therefore use a simple linear regression head and report results in Table 8. Random tokens have trouble to converge and lead to poor results overall. We also experiment with substituting random token sampling with our 2-window strategy instead, either in the first or second frame in the pair. We observe a slight improvement when the second frame is using the 2-window training, but results are still far from the 2-windows + 2-windows baseline, indicating that random tokens intrinsically does not possess the necessary qualities for full-resolution generalization.

Table 8: **Experimenting with uniform random tokens**, instead of multiple windows. In this case, we use a linear head because these window modes are not compatible with a convolutional head.

| Frame 1 masking | Frame 2 masking | Stochasticity | Synthetic EPE↓ |
|---|---|---|---|
| Random (200 tokens) | Random (200 tokens) | 0.0 | 12.50 |
| Random (200 tokens) | Random (200 tokens) | 0.3 | 13.28 |
| Random (200 tokens) | Random (200 tokens) | 0.5 | 3.18 |
| 2 win $10 \times 10$ | Random (200 tokens) | 0.5 | 27.76 |
| Random (200 tokens) | 2 win $10 \times 10$ | 0.5 | 2.96 |
| 2 win $10 \times 10$ | 2 win $10 \times 10$ | 0.5 | **1.63** |

**Results with a linear head.** Table 1 of the main paper shows that Win-Win significantly outperforms a strategy of randomly selecting the tokens. This can be explained by 1) the fact that Win-Win allows to better model local interactions thanks to the window scheme, 2) the convolutional head on top of the backbone which is hard to train with tokens far apart. To verify this, we also evaluate our model with a linear head in the right column of Table 9. With this head, Win-Win still outperforms the random tokens baseline as well as the full-resolution baseline.

Table 9: **Comparison with other approaches using a linear head** on semantic segmentation.

| | # tokens | mIoU↑ conv. head | linear head |
|---|---|---|---|
| Full-res. ViT | 3600 | 63.2 | 61.6 |
| Random tokens | 968 | - | 59.9 |
| **Win-Win (2 win. 22×22)** | 968 | **63.6** | **62.1** |

# D    RESULTS ON MONOCULAR DEPTH ESTIMATION

**Experimental setup.** We also experiment with the monocular task of depth estimation on the NYU v2 dataset (Silberman et al., 2012). The dataset consists of 640×480 images (*i.e.*, 1200 tokens for patches of 16×16 pixels). We use a ConvNeXt prediction head (Liu et al., 2022b) to predict the depth at every pixel. We report the $\delta_1$ metric, *i.e.*, the percentage of pixels such that $\max(d/\hat{d}, \hat{d}/d) < 1.25$ where $d$ is the predicted depth and $\hat{d}$ is the ground-truth one.

**Results.** We report in Table 10 the results for random tokens, full-resolution training with standard ViT or Vit-Det, as well as our approach. 'Random tokens' performs poorly compared to our approach, which allows to better model local interactions and to better train the convolutional head. We perform in the same range as full-resolution while using less than one third of the total number of tokens during training, thus significantly reducing the training time and GPU memory requirement. These results confirm the findings of the other tasks.

Table 10: **Evaluation on monocular depth estimation on NYU v2.**

|  | # tokens | $\delta_1 \uparrow$ |
|---|---|---|
| Random tokens | 392 | 80.8 |
| Full-res. ViT | 1200 | **87.7** |
| Full-res. ViT-Det | 1200 | 86.8 |
| **Win-Win (2-win. 14×14)** | 392 | 87.3 |

# E    TEST-TIME ADJUSTMENT OF THE TEMPERATURE OF THE *softmax* IN THE ATTENTION

The difference of the number of tokens between training with the multi-window scheme and testing at full resolution impacts statistics of the self-attention operation. To showcase this, let us assume a simple case where the probability of attending other tokens is uniform, *e.g.* the query-key product is constant, before RoPE relative positional embedding is applied. In Figure 7, we show attention maps for the token shown in red (top left) once RoPE is applied (bottom left). We compare this to the case of full-resolution inputs (top right) and observe a clear discrepancy, especially when looking at the probability of the red token paying attention to itself. This is due to the significantly larger sets of tokens, leading to an increased divider in the softmax. By setting a well chosen temperature in the softmax, we can obtain an attention map that has a similar behavior in the neighborhood of the red token, and importantly for the self-probability, *i.e.* the weight for attending to itself.

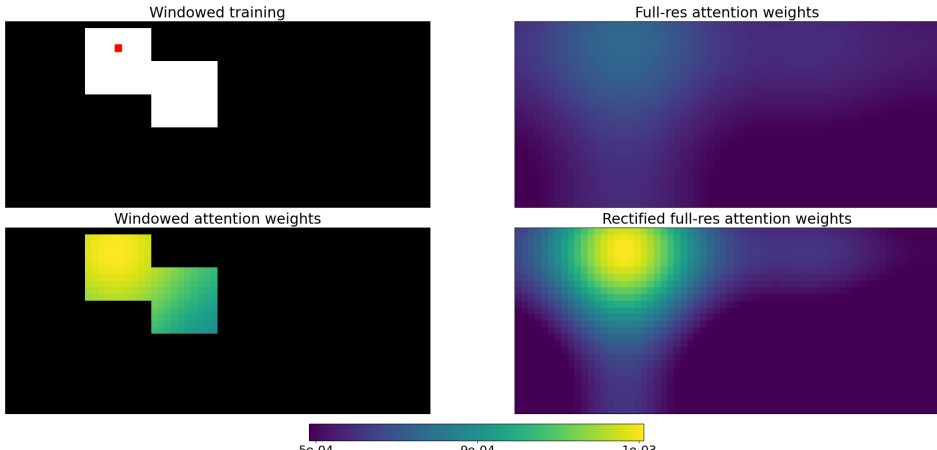

Figure 7: **Attention discrepancy** between train and test. For a given token (red square in top left), the two-window setup used during training (*top left*) will generate attention statistics at train time (*bottom left*) that greatly differ from the attention statistics that happen at test time with full-resolution inputs (*top right*). To correct this, we adapt the temperature of the softmax (*bottom right*).

**Empirical Results.** We set the softmax temperature in self-attention block for full resolution images empirically using *train* images. The full resolution input is used without windows, for both monocular and binocular tasks. For semantic segmentation, we show in Figure 8 (left) the train loss value on the training dataset in full resolution w.r.t. a multiplier of the default temperature. For the optical flow, we plot in Figure 8 (right) the EPE on the train set in full resolution with varying factors as well.

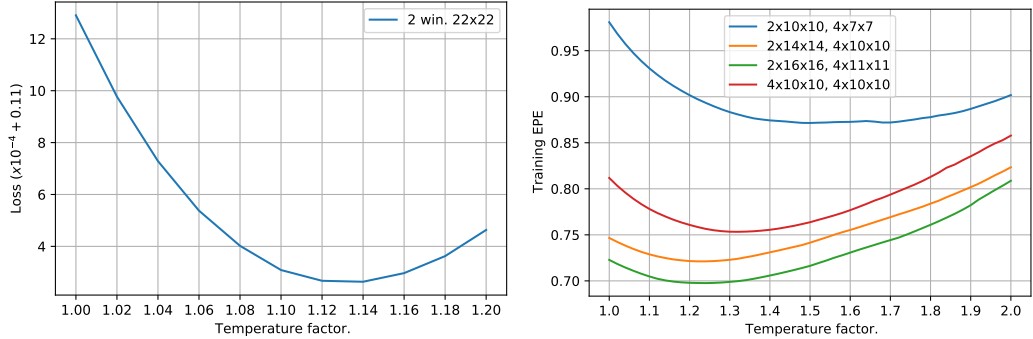

Figure 8: **Impact of temperatures** on the BDD-10k train set for semantic segmentation (*left*) and for optical flow (*right*).

In all our experiments, we clearly observed bell-shaped curves with a minimum value that varies depending on the masking used at train-time. The minima of the curves give us the optimal temperature parameter, that we use at test-time.

**Learning the temperature.** Furthermore, instead of simply validating a single given temperature for all heads of all layers, our results for optical flow estimation on existing leaderboards were obtained by further finetuning the temperatures, *i.e.* learning them while freezing all networks' weights once the network is trained with Win-Win. Because we only finetune a few scalar parameters $\{\tau\}$ in every attention heads, *i.e.* a few hundred parameters, and for very few iterations, this process is fast and the finetuning time (in the order of minutes) is typically negligible compared to the main training. This allows to further reduce the EPE compared to the validated temperature. At the same time, we have observed that the learned temperatures remain close to the validated one.

# F    DETAILED TRAINING SETUPS

## F.1    SEMANTIC SEGMENTATION

We train our models for 200 epochs on the 7,000 training images from the BDD10k dataset (Yu et al., 2020) that has 19 semantic classes. The model is a ViT-Base network with RoPE positional embeddings, initialized with models from (Weinzaepfel et al., 2023). A ConvNeXt head is appended at the end of the vision transformer backbone, following (Bachmann et al., 2022). We use the AdamW (Loshchilov & Hutter, 2019) optimizer, with betas of 0.9 and 0.999, a cosine learning rate schedule with a base learning rate of 0.0001, with two warmup epochs, a weight decay of 0.05 and a learning rate layer decay of 0.75.

To set the *softmax* temperatures after training, we measure the training loss on a subset of 700 images where the forward is run at full resolution, with a temperature swept with steps of 0.02 and chose the one with minimal training loss, as shown in Figure 8 (left).

## F.2    OPTICAL FLOW

We follow the CroCo-Flow architecture (Weinzaepfel et al., 2023): each image is embedded using a ViT-Base backbone with RoPE positional embeddings. A binocular decoder then uses a transformer decoder architecture to output one token for each patch of the first image. A convolutional head is used for prediction the flow along the x axis, along the y axis and an uncertainty measure. It is based on DPT (Ranftl et al., 2021) that are decoupled by first learning a fully-connected layer for each of these 3 predictions, before a shared DPT is run on these features. We use the pre-trained weights with cross-view completion (Weinzaepfel et al., 2022; 2023) for the ViT encoder and the decoder. We optimize a Laplacian loss, *i.e.*, that minimizes the L1 loss while taking into account the predicted uncertainty (Kendall et al., 2018). We use the AdamW (Loshchilov & Hutter, 2019) optimizer, with betas of 0.9 and 0.95, a cosine learning rate schedule with a base learning rate of 0.0001, with one warmup epoch, a weight decay of 0.05 and a minimum learning rate of $10^{-6}$. We train the model

for 100 epochs. When submitting to the MPI-Sintel test set, we finetune the model with the addition of the TartanAir dataset (Wang et al., 2020b) for 15 epochs.

To set the *softmax* temperatures after training, we measure the training EPE on the Sintel train set, where the forward is run at full resolution with a temperature between 1.0 and 2.0 with a step of 0.02 and chose the one with minimal value, as shown in Figure 8 (right).

## G  VARIANCE OF THE RUNS

We report the standard deviation (*std*) over 3 runs of several key experiments here. This study is limited to a select few experiments, to limit computation costs.

For semantic segmentation, our Win-Win training strategy with two windows of size $22 \times 22$ tokens has a *std* of 0.28. This is why we reported only decimal with 1 digit, and do not consider runs 0.1 above the baseline in Table 1 right of the main paper as performing better. For comparison, when training on crops of $32 \times 32$ tokens and testing with tiles overlapping with a ratio of 50%, the standard deviation is 0.37 on 3 runs.

For optical flow, when training with 2 windows of size $16 \times 16$ in the first image and 4 windows of size $11 \times 11$ in the second, we obtain a standard deviation of 0.07.

