# OpenReview forum: "Win-Win: Training High-Resolution Vision Transformers from Two Windows"
_ICLR.cc/2024/Conference — ICLR 2024 poster_

### Official Review · Reviewer_aJaX · 2023-10-30

**Soundness:** 3 good
**Presentation:** 3 good
**Contribution:** 2 fair
**Rating:** 5
**Confidence:** 4

**Summary:**

In this paper, the authors tackle the challenges in training high-resolution vision transformers. ViTs have quadratic complexity with respect to the input resolution. Hence, reducing the number of tokens during training is helpful. To this end, the authors proposed Win-Win, an approach that optimizes token selection during training, balancing local and long-range interactions, and is adaptable to various computer vision tasks, including semantic segmentation and optical flow estimation. The Win-Win approach reduces training complexity by using a subset of tokens, denoted as P', during training. This involves masking (discarding) the remaining tokens. This reduces the training complexity from O(|P|^2) to O(|P'|^2), where the size of P' is much smaller than P.

**Strengths:**

- The paper is well-written and easy to follow.

- The proposed idea is interesting. The authors demonstrated that partial supervision at training time can also work for pixelwise prediction tasks in ViT.

- Win-Win leads to measured training speedup for high-resolution ViTs.

- Validation results on BDD100K and MPI-Sintel indicates that partial supervision at training time does not significantly impact the performance at test time.

**Weaknesses:**

- The evaluation appears to be somewhat constrained. High-resolution Vision Transformers (ViTs) possess a wide range of potential applications, including object detection, semantic segmentation, instance segmentation, and serving as a backbone in diffusion models. However, the authors have restricted their analysis to just two representative applications and a limited set of datasets. This limitation raises questions about the general applicability of the proposed method. Specifically, I am keen to ascertain the viability of applying this method to object detection and diffusion models.

- The ablation study regarding window selection is currently lacking completeness. Notably, an essential baseline - random and non-contiguous window selection - has been omitted for semantic segmentation tasks. I am curious whether Win-Win works better than random masking in single-frame applications, though Table 8 demonstrated its effectiveness for optical flow prediction.

- The paper brings to mind GridMask, a well-known data augmentation technique widely applicable in various vision-related tasks. It would be beneficial if the author could incorporate a discussion on GridMask, highlighting the similarities and differences with the Win-Win method. Furthermore, the paper lacks any discussion regarding inference acceleration techniques for Vision Transformers (ViTs) through token pruning, such as DynamicViT, A-ViT, IA-RED^2, and SparseViT. Incorporating a brief comparison or analysis of Win-Win in the context of these methods would enhance the completeness of the paper.

**Questions:**

Please respond to my concerns in the "Weaknesses" section.

---

> ### Author Response · Authors · 2023-11-21
> **Author response - part 1 of 2**
>
> We sincerely thank the reviewer for the relevant remarks and points raised. We answer in the following about the applicability, ablations studies, missing references, and about the relationship to token pruning. The paper was updated accordingly.
>
> > *The evaluation appears to be somewhat constrained. High-resolution Vision Transformers (ViTs) possess a wide range of potential applications, including object detection, semantic segmentation, instance segmentation, and serving as a backbone in diffusion models. However, the authors have restricted their analysis to just two representative applications and a limited set of datasets. This limitation raises questions about the general applicability of the proposed method. Specifically, I am keen to ascertain the viability of applying this method to object detection and diffusion models.*
>
> Thanks for your suggestion. In this paper we focus on pixelwise task. This is because for other tasks, hierarchical backbones already offer a viable alternative to train at high resolution. For instance they have been shown to work for tasks like object detection. However these hierarchical backbones have inherent limitations when considering pixelwise tasks, in particular because maintaining the input resolution is desirable.
>
> We do understand and agree with the request for more tasks. To provide that, we experimented with the task of monocular depth estimation on the NYU v2 dataset. While it was not specifically requested, we believe it was the best candidate because a) it is pixelwise as discussed, and b) it provides reasonably high-resolution images, which is also necessary for our method to be relevant.
>
> Below, we report preliminary results with the delta-1 (higher is better) metric on this dataset for full-resolution training, ViT-Det, random tokens and our approach. These results confirm the previous finding of the paper. We perform on par with full-resolution training while being 3 to 4 times faster to train. These results are mentioned in Section 4.1 and detailed in Appendix D and Table 10 of the revised paper.
>
> | NYU Depth (delta 1)                      |         |
> |------------------------------------------|---------|
> | Full-res ViT      (1200 tokens)          |   87.7  |
> | Full-res ViT-Det  (1200 tokens)          |   86.8  |
> | Random tokens (392 tokens)               |   80.8  |
> | **Win-Win (2-win. 14x14) (392 tokens)** |   87.3  |
>
> > *The ablation study regarding window selection is currently lacking completeness. Notably, an essential baseline - random and non-contiguous window selection - has been omitted for semantic segmentation tasks. I am curious whether Win-Win works better than random masking in single-frame applications, though Table 8 demonstrated its effectiveness for optical flow prediction.*
>
> Great point. Thanks. We initially did not run this ablation as masking random regions is not compatible with typical convolutional heads that are run on top (e.g. mentioned at the end of page 2).
> On semantic segmentation, when running our method with 968 randomly sampled patches, i.e., with a similar number of patches as Win-Win, we obtain a mIoU performance of 59.9, compared to 63.6 with our approach.
>
> To verify that this is not simply due to the fact that training the convolutional head with a larger receptive field is better, we also try with a simple linear head. In this case, we obtain 59.9 for randomly selected patches, 61.6 with full-resolution training and 62.1 with our proposed Win-Win approach.
>
> These results further highlight the benefit of our approach with structured masking.
> Intuitively, using structured masking allows to model both local and global interactions between patches, while training significantly faster than the full-resolution counterpart. We added this baseline to Table 4 in the revised paper, and the results with a linear head in Appendix C and Table 9.
>
> |                                    | conv. head | linear head |
> |------------------------------------|------------|-------------|
> | Random tokens (968 tokens)         | 59.9       | 59.9        |
> | Win-Win (968 tokens, 2 win. 22x22) | **63.6**   | **62.1**    |
> | Full-resolution (3600 tokens)      | 63.2       | 61.6        |

---

> ### Author Response · Authors · 2023-11-21
> **Author response - part 2 of 2**
>
> > *The paper brings to mind GridMask, a well-known data augmentation technique widely applicable in various vision-related tasks. It would be beneficial if the author could incorporate a discussion on GridMask, highlighting the similarities and differences with the Win-Win method. Furthermore, the paper lacks any discussion regarding inference acceleration techniques for Vision Transformers (ViTs) through token pruning, such as DynamicViT, A-ViT, IA-RED^2, and SparseViT. Incorporating a brief comparison or analysis of Win-Win in the context of these methods would enhance the completeness of the paper.*
>
> Thanks a lot for bringing the GridMask reference which we had missed. In GridMask, the idea is to color some squared areas in black, to act as a data augmentation for semantic tasks: the model should implicitly leverage context to implicitly fill-in the missing information, similar to MAE for pre-training.
>
> The small gain we observe for our approach compared to full-resolution training may be explained by the fact that it also performs a similar augmentation. We thus add and discuss this insight in the revised paper (at the end of page 6).
>
> We also note that our method has key differences with it:
> * a) We mainly aim at efficient training despite the quadratic complexity of attention: we completely remove patches, similarly to MAE, rather than simply changing their color. We do not wish to claim quantitative performance gains from using it as a data augmentation tool, as it is less clear what to expect from this augmentation when considering more geometric tasks.
> * b) An other difference is that we use a structured masking where large areas are completely removed: it is thus harder to implicitly fill-in the blank in these regions
> * c) We extend it to the case of binocular task with a smart masking strategy in the second image, guided by flow. We show that in this case, naively masking both images is sub-optimal.
>
> Regarding the token pruning strategies, we mention them in the second point of the second paragraph of the introduction, where we added the suggested references, thanks. They are indeed relevant, but these token pruning strategies are most of the time not applicable to pixelwise prediction tasks and rather to image-level or object-level tasks, for which reducing the number of tokens is not harmful. In contrast, with pixelwise prediction tasks, it is natural to keep the input resolution intact.

---

> > ### Comment · Reviewer_aJaX · 2023-11-22
> >
> > Thank you for providing such a thorough response. I appreciate the detailed information and find that most of my questions have been adequately addressed. After careful consideration, I am inclined to revise my score to 5. Although the proposed method is intriguing, I remain uncertain about its broad applicability. It is unclear whether other crucial applications, such as object detection could benefit from Win-Win to reduce their training time. Besides, the number of datasets used for segmentation is also too small. Based on the comments above, I find it challenging to further increase the score to 6.

---

### Official Review · Reviewer_yL3d · 2023-10-31

**Soundness:** 3 good
**Presentation:** 2 fair
**Contribution:** 2 fair
**Rating:** 6
**Confidence:** 3

**Summary:**

The paper addresses the challenge of training vision transformers for high-resolution pixelwise tasks, which is computationally expensive. The authors propose a novel strategy called Win-Win, which involves masking out most of the high-resolution inputs during training and keeping only a few random windows~(2 windows are enough). The model can learn local and global interactions efficiently. The proposed strategy is faster to train, straightforward to use at test time, and achieves comparable performance on semantic segmentation and optical flow tasks.

**Strengths:**

1. Easy to implement: The method is simple and easy to integrate into the dense prediction tasks like semantic segmentation and binocular tasks like optical flow estimation.
2. Efficient Training: The Win-Win strategy reduces training time by a factor of 4 compared to full-resolution networks. It achieves this by focusing on random windows instead of processing the entire high-resolution input.

**Weaknesses:**

1. Lack of Comparative Analysis: It does not provide a comprehensive comparative analysis with a wide range of existing methods for training high-resolution vision transformers.
2. Lack of Novelty: Masking out image patches is not a new approach in the literature, and it is simple data augmentation tuning to mask out most of them. I highly recommend the authors to do in-depth exploration and analysis of this method.

**Questions:**

None

---

> ### Author Response · Authors · 2023-11-21
> **Author response - part 1 of 2**
>
> We gladly thank the reviewer for the constructive feedback. We answer the points regarding the lack of comparisons with other ViT architectures, and the novelty.
>
> > *Lack of Comparative Analysis: It does not provide a comprehensive comparative analysis with a wide range of existing methods for training high-resolution vision transformers.*
>
> Most works on fast vision transformers do not tackle pixelwise prediction tasks. In particular, many of them rely on pooling or downsampling tokens, which definitely harm pixelwise predictions. For such problems, maintaining the input resolution throughout is an intuitive solution. We provide a solution to do that while keeping the computational budget constrained. We do welcome any specific method suggestion to compare with.
>
> > *Lack of Novelty: Masking out image patches is not a new approach in the literature, and it is simple data augmentation tuning to mask out most of them. I highly recommend the authors to do in-depth exploration and analysis of this method.*
>
> Masking out image patches has been extensively studied in the context of pre-training image representation, following the seminal work of MAE [He et al, 2022] or as a form of data augmentation e.g with MSN [Assran et al, 2022]. These types of masking are quite different from ours.
> * a) In our case, the targets are aligned with the *visible* patches, contrary to MAE where they are aligned with the *masked* patches, or to MSN where the targets are *invariant* to the masking. This is because these methods tackle pre-training and not downstream tasks.
> * b) Our proposed method goes beyond a naive random masking of patches: we use a **structured masking strategy** where we mask out all tokens except two windows. This allows to model both local interaction and global interaction that are both capital in downstream performances. In the case of multiple input views, we show that guiding the window selection using relevant signal is key to preserve performance, which does not happen with random masking.
> * c) Our purpose is not to improve quantitative performance, which is what typical `simple data augmentation tuning' would achieve. It is to enable efficient training at high resolutions; we believe that this claim is well substantiated (see e.g. Figure 1). The masking is not a data augmentation here, it is a way to control the computational budget.
>
> We actually did compare to random tokens for optical flow in Table 8, and we now also provide a comparison on semantic segmentation where with 968 random tokens, we obtain 59.9 mIoU instead of 63.6 with our structured masking strategy. This loss of performance with random masking is not only due to the fact that it is harder to train convolutional head on top, as we still obtain a significant gain up to 62.1 compared to 59.9 with random masking.
>
> Our results demonstrate that a structured masking strategy consistently outperforms random masking, for both monocular and binocular tasks. Extending a masking strategy to binocular tasks like optical flow is also novel.We gladly thank the reviewer for the constructive feedback. We answer the points regarding the lack of comparisons with other ViT architectures, and the novelty.
>
> > *Lack of Comparative Analysis: It does not provide a comprehensive comparative analysis with a wide range of existing methods for training high-resolution vision transformers.*
>
> Most works on fast vision transformers do not tackle pixelwise prediction tasks. In particular, many of them rely on pooling or downsampling tokens, which definitely harm pixelwise predictions. For such problems, maintaining the input resolution throughout is an intuitive solution. We provide a solution to do that while keeping the computational budget constrained. We do welcome any specific method suggestion to compare with.

---

> ### Author Response · Authors · 2023-11-21
> **Author response - part 2 of 2**
>
> > *Lack of Novelty: Masking out image patches is not a new approach in the literature, and it is simple data augmentation tuning to mask out most of them. I highly recommend the authors to do in-depth exploration and analysis of this method.*
>
> Masking out image patches has been extensively studied in the context of pre-training image representation, following the seminal work of MAE [He et al, 2022] or as a form of data augmentation e.g with MSN [Assran et al, 2022]. These types of masking are quite different from ours.
> * a) In our case, the targets are aligned with the *visible* patches, contrary to MAE where they are aligned with the *masked* patches, or to MSN where the targets are *invariant* to the masking. This is because these methods tackle pre-training and not downstream tasks.
> * b) Our proposed method goes beyond a naive random masking of patches: we use a **structured masking strategy** where we mask out all tokens except two windows. This allows to model both local interaction and global interaction that are both capital in downstream performances. In the case of multiple input views, we show that guiding the window selection using relevant signal is key to preserve performance, which does not happen with random masking.
> * c) Our purpose is not to improve quantitative performance, which is what typical `simple data augmentation tuning' would achieve. It is to enable efficient training at high resolutions; we believe that this claim is well substantiated (see e.g. Figure 1). The masking is not a data augmentation here, it is a way to control the computational budget.
>
> We actually did compare to random tokens for optical flow in Table 8, and we now also provide a comparison on semantic segmentation where with 968 random tokens, we obtain 59.9 mIoU instead of 63.6 with our structured masking strategy. This loss of performance with random masking is not only due to the fact that it is harder to train convolutional head on top, as we still obtain a significant gain up to 62.1 compared to 59.9 with random masking.
>
> Our results demonstrate that a structured masking strategy consistently outperforms random masking, for both monocular and binocular tasks. Extending a masking strategy to binocular tasks like optical flow is also novel.

---

> > ### Comment · Reviewer_yL3d · 2023-11-23
> >
> > Thanks for the thorough response.  Most of my questions of comparative analysis to other similar existing works~(like grid-mask and the improved vision-transformers) can be addressed from the other comments and author’s reply.
> >
> > For the novelty part, the author emphasizes that the core contribution lies in `certainty~(or structured)` masking strategies.   I am still interested in this insight into this simple training strategy. If it is effective for various tasks , like det,seg,opt-flow), it means there may be a theoretical guarantee or a deeper mechanism.
> > Based on above comments， I will change my rating to 6.

---

### Official Review · Reviewer_Jtip · 2023-11-01

**Soundness:** 3 good
**Presentation:** 3 good
**Contribution:** 3 good
**Rating:** 6
**Confidence:** 3

**Summary:**

The authors introduce a novel strategy for efficient high-resolution vision transformer training and inference. This approach involves masking most high-resolution inputs during training, retaining only N random windows. It enables the model to learn local and global token interactions, allowing direct processing of high-resolution input during testing without special techniques. Notably, this method is four times faster to train than full-resolution networks and straightforward to implement during testing. The authors apply this approach to semantic segmentation and optical flow tasks, achieving state-of-the-art performance on the Spring benchmark with significantly reduced inference times

**Strengths:**

Strength:
1. The proposed model provides competitive performance while reducing the training time
2. The proposed model can be generalized and applied to various tasks like segmentation and binocular task of optical flow.
3. It is easy to apply the proposed strategy.

**Weaknesses:**

1. The paper presents semantic segmentation result for a single train and test resolution (1280x720). Does the performance hold for the solution exceeding 1280x720?
2. Win-Win is better than ViT-Det by a mere 0.3%, suggesting a marginal enhancement. Can 0.3% deemed as a substantial improvement?

**Questions:**

Please refer to the weakness section.

---

> ### Author Response · Authors · 2023-11-21
> **Author response**
>
> Thanks for your constructive and positive feedback. We now address the concerns raised, in particular regarding the scalability to higher image resolutions, and the substantiality of some improvements.
>
> > *The paper presents semantic segmentation result for a single train and test resolution (1280x720). Does the performance hold for the solution exceeding 1280x720?*
>
> Yes, it would work at higher resolution. As an example, our optical flow experiments on the Spring benchmark are performed on **1920x1080** images.
>
> We have rephrased the beginning of the last paragraph of Section 4.2 where the results on Spring are presented to insist more on how critical it becomes to reduce training and inference costs when dealing with such large resolutions.
>
> > *Win-Win is better than ViT-Det by a mere 0.3%, suggesting a marginal enhancement. Can 0.3% deemed as a substantial improvement?*
>
> It is indeed a rather limited improvement, but an improvement nonetheless. Most importantly, we do not seek improved performance and do not wish to claim it, we have made sure that the paper is clear in that respect. We do wish to claim that Win-Win is efficient: it is **3 times faster** to train than ViT-Det (Figure 1) and **4 times faster** to train than ViT. It is also conceptually simple, and does not restrict nor modify the vanilla transformer architecture. Additionally, our contribution of using structured masking is orthogonal to attention approximation and thus open new ways of tackling the problem. We believe this makes the overall contribution significant.

---

> > ### Comment · Reviewer_Jtip · 2023-11-22
> >
> > Thank you for your reply. I will maintain the existing score

---

### Official Review · Reviewer_wXRr · 2023-11-02

**Soundness:** 3 good
**Presentation:** 3 good
**Contribution:** 3 good
**Rating:** 6
**Confidence:** 5

**Summary:**

The paper proposes a ViT training strategy that allows plain-ViT to train with high-resolution images with masking, and inference with high-resolution images with a single forward. Specifically, it proposes to keep only tokens from two randomly selected non-overlapping windows at training time for ViT to process. The paper empirically shows the proposed way (win-win) of training ViT achieves better training and inference cost trade-off with comparable performance of training on full-resolution. Experiments are conducted on both monocular task of semantic segmentation on BDD-100k and binocular task of option flow estimation.

**Strengths:**

The proposes method enables a simple single forward inference process on high-resolution images without performance drop. In comparison to existing approach, most requires extra effort of aligning train and test resolution difference, e.g. aggregating predictions from multiple small patches.

The ablation study regarding the window generation strategy is quite extensive, including various ways of choosing windows, how many windows, window size, square or non-square window, etc.

The paper also extends the approach to optical flow, which requires more specific design of choosing window as two images involved, and the experiments results listed shows its potentials compared to other methods.

**Weaknesses:**

While the extensive experiments show the two window strategy is the best one with its simple and good performance, some analysis of such strategy/experiment results is missing. For example, in Table 1 right, why using 1021 tokens will drop 0.6 performance to using 1009 is not clear.

The results from Table 3 indicates the select of window strategy affects a lot for optical flow task, which suggests difficulties of generating to other data/task (search of window is needed).

**Questions:**

1.	For results in Table 1, row 1 (968 tokens) compare with row 2 (988), using non-square window has a large drop of 1 mIOU, do authors have any insights on that? And for row 5 (1021 tokens), why a 0.6 mIOU drop compared to 1009 tokens, it is because of min window size?
2.	Table  3 shows larger variance when using different window strategy compared to table 1, any insights for this?
3.	From Table 3, the best setting is 2 win. 10x10, 4 win. 7x7, then as state ‘using the best settings found previously), to eval on MPI-Sintel’, however, the setting used have different window size, how did the author choose the window size on MPI-Sintel? Does this mean each time, a search of window size is needed when applying to different data?
4.	In table 5, compared to CroCoFlow, which is also the paper’s base method, it has a performance drop, authors states using different backbone (vit-l v.s. vit-base), the comparison of using the same backbone is missing.

---

> ### Author Response · Authors · 2023-11-21
> **Author response - part 1 of 2**
>
> We were glad to read the positive and constructive remarks. We answer below the questions raised, concerning the number of tokens for comparisons, non-square windows, metrics variance, choosing the number of visible tokens, and comparisons to CroCo-Flow.
>
> ### Weaknesses:
> > *While the extensive experiments show the two window strategy is the best one with its simple and good performance, some analysis of such strategy/experiment results is missing. For example, in Table 1 right, why using 1021 tokens will drop 0.6 performance to using 1009 is not clear.*
>
> Thanks for your question. Our paper shows that the number of tokens is not what matters the most, and that the structure of the masking has more impact. In Table 1 right, we aim at comparing different rectangle shapes for the two windows. We make this comparison while keeping the number of tokens approximately constant. This table mainly shows that having the two windows of the same size performs better than having two windows of different sizes, which is also the simplest strategy. One hypothesis is that this window scheme with the same size allows to better model local interactions compared to an asymmetric one.
>
> > *The results from Table 3 indicates the select of window strategy affects a lot for optical flow task, which suggests difficulties of generating to other data/task (search of window is needed).*
>
> The average endpoint error (EPE) metric for optical flow corresponds to the difference in pixels between the ground-truth and estimated motions. In other words, the maximal difference range is less than 1 pixel, which remains rather small.

---

> ### Author Response · Authors · 2023-11-21
> **Author response - part 2 of 2**
>
> ### Questions:
>
> > *For results in Table 1, row 1 (968 tokens) compare with row 2 (988), using non-square window has a large drop of 1 mIOU, do authors have any insights on that? And for row 5 (1021 tokens), why a 0.6 mIOU drop compared to 1009 tokens, it is because of min window size?*
>
> This could be a possible explanation. However, the main message from Table 1 is that, to decide the size of the two windows, choosing two squared windows of the same size is simplest and efficient. One explanation could be that this allows to maximize the number of training instances and the size for local interactions.
>
> > *Table 3 shows larger variance when using different window strategy compared to table 1, any insights for this?*
>
> The metrics are significantly different. For optical flow, it is an average error in pixels, compared to the standard mean intersection-over-union for semantic segmentation, where changes in scores have an effect on the metric only if this changes the relative ordering of the score classes.
>
> > *From Table 3, the best setting is 2 win. 10x10, 4 win. 7x7, then as state ‘using the best settings found previously), to eval on MPI-Sintel’, however, the setting used have different window size, how did the author choose the window size on MPI-Sintel? Does this mean each time, a search of window size is needed when applying to different data?*
>
> Good point, thanks. We did indeed increase the number of tokens, which differs from the setting in the ablations on the synthetic dataset and the sentence was indeed lacking clarity. We did keep the rest of the configuration: when increasing the number of tokens, we keep the number and shapes of the windows and thus simply increase their sizes. We report below the ablation when increasing the number of tokens with around 200, 300, 400 and 500 tokens. Note that 500 tokens corresponding roughly to one quarter of the total number of patches for the full-resolution images for MPI-Sintel, a similar ratio as for semantic segmentation. This number of tokens is also close to the one used by CroCo-Flow (384x320 crops, which represents 480 tokens). We added these results to the main paper in Section 4.2 and Table 4 left.
>
> | window settings                   | (#ntokens img1/img2) | EPE val |
> |-----------------------------------|----------------------|---------|
> | 2 win. 10x10 -> 4 win. 7x7        | (200/196)            | 2.00    |
> | 2 win. 12x12 -> 4 win. 9x8        | (288/288)            | 1.96    |
> | 2 win. 14x14 -> 4 win. 10x10      | (392/400)            | 1.83    |
> | **2 win. 16x16 -> 4 win. 11x11    | (512/484)            | **1.67**|
>
> > *In table 5, compared to CroCoFlow, which is also the paper’s base method, it has a performance drop, authors states using different backbone (vit-l v.s. vit-base), the comparison of using the same backbone is missing.*
>
> The results from Table 5 are from the official leaderboard. Results for Croco-Flow are with a ViT-Large backbone while we limited the scope of this paper to ViT-Base backbones to control our computational budget. We cannot have the results of CroCo-Flow with a ViT-Base backbone as this benchmarks uses an official leaderboard.
>
> However, looking at values reported in the CroCo-Flow paper, we observe a difference of 0.08 EPE on MPI-Sintel clean between the Large and the Base backbone on the validation set, and both backbones are on par on MPI-Sintel final.
>
> Our Win-Win method is only 0.06 EPE below CroCo-Flow (Large) backbone on the official leaderboard on the clean rendering pass, and is performing 0.10 better on the final pass. Additionally, note that CroCo-Flow (with the large overlap of 0.9 between tiles at inference time used for the submission to the official leaderboard) is 20 times slower at inference when using the same backbone size!
>
> To summarize, our main contribution is not about improving the final performance. Rather, we provide a simple method to train fast at high resolution and with a straightforward inference, compared to training on crops and using tiling at inference, or training on full-resolution images (see Figure 1 right). Our approach does maintain the task performance.

---

> > ### Comment · Reviewer_wXRr · 2023-11-23
> > **thanks for the response**
> >
> > Thank authors for the detailed response, most of my questions are resolved. I will keep my rating as 6.

---

### Author Response · Authors · 2023-11-21
**Authors general response to reviews**

We thank the reviewers for their insightful feedback. We gratefully note that reviewers have found the method "*novel*" (R.Jtip, R.yL3d), "*efficient*" (R.yL3d) and with "*a wide range of potential applications*" (R.yL3d, R.Jtip). We were also pleased to read that all reviewers found the method "*easy to apply/implement*" (R.Jtip, R.yL3d), "*leads to training speedup*" (R.aJax) "*four times faster to train than full-resolution networks and straightforward to implement during testing*" (R.Jtip), while "*not significantly impacting the performance*" (R.aJax).

We answer the concerns raised by each reviewer in the individual responses below. We hope that our responses will provide clarifications and we look forward for feedback. We stay available if any question ever remain.

We have also revised the paper accordingly, while coloring the changes in blue for easier readability.

---

### Meta-Review · Area_Chair_cUhd · 2023-12-06

**Metareview:**

The paper proposes a simple method for speeding up training of vision transformers for pixel-wise prediction tasks. Namely, at each training step the model is applied only to a subset of tokens from the image, selected in the shape of several (in the end, two) square windows. The method is evaluated on semantic segmentation and optical flow estimation and yields similar results to a standard vision transformer while being 3-4x times faster to train.

After the rebuttal and discussion, the reviewers still have a somewhat mixed opinion of the paper leaning towards positive. The key pros and cons are as follows.

Pros:
1. Simple and effective method
2. Good training speedups
3. A fairly extensive ablation study
4. Application not only to single-input-image tasks like semantic segmentation, but also to tasks taking two images as input, namely optical flow

Cons:
1. Somewhat limited evaluation - would be nice to have more semantic segmentation datasets and more tasks - e.g. detection. (depth estimation results reported in the rebuttal go in this direction and are nice to have)
2. Lack of comparison to various ViT variants

Overall, while it would be nice to extend the experiments further, the paper seems methodologically convincing and interesting as it stands. I therefore recommend acceptance. But I encourage the authors to take the reviewers' feedback into account and improve the paper for the final version.

**Justification For Why Not Higher Score:**

1. Somewhat limited evaluation - would be nice to have more semantic segmentation datasets and more tasks - e.g. detection. (depth estimation results reported in the rebuttal go in this direction and are nice to have)
2. Lack of comparison to various ViT variants

**Justification For Why Not Lower Score:**

1. Simple and effective method
2. Good training speedups
3. A fairly extensive ablation study
4. Application not only to single-input-image tasks like semantic segmentation, but also to tasks taking two images as input, namely optical flow

---

### Decision · Program_Chairs · 2024-01-16

Accept (poster)